# Mass Spectrometry-Based Peptide Profiling of Haemolymph from *Pterostichus melas* Exposed to Pendimethalin Herbicide

**DOI:** 10.3390/molecules27144645

**Published:** 2022-07-21

**Authors:** Donatella Aiello, Anita Giglio, Federica Talarico, Maria Luigia Vommaro, Antonio Tagarelli, Anna Napoli

**Affiliations:** 1Department of Chemistry and Chemical Technologies, University of Calabria, 87036 Arcavacata di Rende, Italy; donatella.aiello@unical.it (D.A.); antonio.tagarelli@unical.it (A.T.); 2Department of Biology, Ecology and Earth Science, University of Calabria, 87036 Arcavacata di Rende, Italy; anita.giglio@unical.it (A.G.); federica.talarico@unical.it (F.T.); marialuigia.vommaro@unical.it (M.L.V.)

**Keywords:** Carabid beetles, pendimethalin, haemolymph, MALDI mass spectrometry, chemometric analysis, elimination of toxicants

## Abstract

Pendimethalin-based herbicides are used worldwide for pre-emergence selective control of annual grasses and weeds in croplands. The endurance of herbicides residues in the environment has an impact on the soil biodiversity and fertility, also affecting non-target species, including terrestrial invertebrates. Carabid beetles are known as natural pest control agents in the soil food web of agroecosystems, and feed on invertebrates and weed seeds. Here, a mass spectrometry untargeted profiling of haemolymph is used to investigate *Pterostichus melas* metabolic response after to pendimethalin-based herbicide exposure. Mass spectrometric data are examined with statistical approaches, such as principal component analysis, for possible correlation with biological effects. Those signals with high correlation are submitted to tandem mass spectrometry to identify the associated biomarker. The time course exposure showed many interesting findings, including a significant downregulation of related to immune and defense peptides (M-lycotoxin-Ls4a, Peptide hormone 1, Paralytic peptide 2, and Serine protease inhibitor 2). Overall, the observed peptide deregulations concur with the general mechanism of uptake and elimination of toxicants reported for Arthropods.

## 1. Introduction

Dinitroaniline herbicides, including trifluralin, ethalfluralin, oryzalin, and pendimethalin, have been used worldwide for pre-emergence selective control of annual grasses and weeds in croplands. Pendimethalin ([N-(1-ethylpropyl)-2,6-dinitro-3,4-xylidine], PND) acts on the polymerization of microtubules, interfering with cell division (mitosis) and, thus, inhibiting the development of roots and shoots in seedlings [1]. However, the endurance of PND residues in the environment have an impact on the growth the soil bacterial [2] and fungal [3], consequently altering the soil biodiversity and fertility [4]. Its sublethal effects have been documented in several non-target species including terrestrial invertebrates, such as springtails, earthworms, pill bugs [5], wasps [6], and beetles [7], performing ecosystem services as decomposers, parasitoids, and pest predators, respectively. Carabid beetles are known as natural pest control agents in the soil food web of agroecosystems [8] to feed on invertebrates (aphids, beetles, lepidopterans, slugs, and dipterans) [9] and weed seeds [10]. However, the routine application of agrochemicals causes the exposure of all beneficial insects inhabiting agricultural landscapes, including carabids in soil trophic web, through direct contact with the sprayed products or by eating contaminated food. Currently, laboratory, field, and semi-field studies have been carried out to assess the detrimental effects of pesticides at the organism level [11], and the resulting impact on diversity and abundance of carabid species [12]. Some studies have also supplied evidence that the exposure to herbicides causes mortality or sublethal effects in carabids [13,14]. Metabolomics provide interesting information for the small molecules, and strongly linked to the phenotype [15]. One of the most established protocols in metabolomics is the metabolic profiling biological fluids, which are easy to collect and store, and contain a wide range of molecules that can be strongly affected by physiological disorder including nutrition or disease [16]. Insect haemolymph plasma is a complex mixture of inorganic ions and cations, amino acids, proteins (storage and lipid transport proteins), lipids, carbohydrates, and their degradation products [17]. It is primarily responsible for supplying nutrients and transferring metabolic wastes to maintain homeostasis. The haemolymph plays a central role in the host immune defense processes and provides constant carriage of haemocytes involved in cellular immune responses, i.e., pathogen recognition, phagocytosis, melanization, nodulation, and encapsulation, and humoral effectors including antimicrobial peptides, enzymes that regulates coagulation and melanization, and reactive oxygen (ROS) and nitrogen (RNS) species [18]. Therefore, this fluid represents an opportunity to access to metabolic pool and to collect evidence of metabolic disorders along time-periods [19]. Mass spectrometry profiling (MS) is a technique commonly used to characterize complex matrices and biological fluids [20,21]. Matrix assisted laser desorption/ionization-time of flight mass spectrometry (MALDI-TOF MS) has become increasingly valuable for insect taxonomy [22,23], and also for monitoring significant haemolymph molecular changes during developmental stages [24] or infections [25]. This technique is used to analyze peptides and small proteins located in cells and tissues [26,27] or released into the haemolymph, such as hormones, to regulate diuresis, heartbeat, ecdysis [28], and antimicrobial peptides involved in immune responses [29]. In addition, it is applied to evaluate alterations in protein distribution and expression effects under the agrochemical exposure [30,31,32]. A high resolution of MS data and accurate structure characterization can lead to study a complete set of low molecular weight species in a biological sample drawing a specific fingerprint, time related and ambient associated [33]. This approach has the advantage to highlight both known and unknown metabolites. Identification of metabolites, however, is necessary to draw biological conclusions from untargeted MS data. This step is generally performed by searching the experimental MS/MS data through databases available to the public for free. MS-untargeted profiling of haemolymph is a useful tool to investigate the metabolic response to pendimethalin-based herbicide exposure. *P melas*, a eurytopic and thermophilus clay soil species inhabiting pastures, open forests, and forest edges in Central and Southern Europe, was selected as a case study. The carabid beetle *Pterostichus melas* italicus Dejean, 1828 (Coleoptera, Carabidae) is a generalist predator of pests including aphids, lepidopterans, and dipterans [34]. The exposure effect is tested in vivo on *P. melas* males and females in the reproductive phase of their life cycle to simulate the field exposure in their main period of activity. A recommended field rate (4 L per ha, 330 g/L^−1^ of active ingredient) is used to evaluate the variability of responses in 21 days, corresponding to the half-life of PND [1,4]. The MS data (all signals) are examined with statistical approaches such as principal component analysis (PCA) for possible correlation with the observed biological effects. Those signals with high correlation are submitted to MS/MS experiments to identify the associated biomarker. Finally, relevant protein–protein interactions network analysis is performed by the STRING [35] database (Search Tool for the Retrieval of Interacting Genes/Proteins).

## 2. Results and Discussion

For metabolic profiling, haemolymph samples from both males and females of *P. melas* control and PND-treated were collected at 2, 7, and 21 days after the initial exposure to PND.

### 2.1. MALDI MS Metabolites Identification

The haemolymph from males and females of *P. melas* was analyzed by MS, in both control and PND-treated specimens. MS spectra revealed different ion species with different intensity reflecting the individual variability (sex or response to treatment). The identities of some of the detected molecules were assigned through proteomics-generated data sets. A total of twenty-three peptides are identified in a single spectrum (Table 1). Although, the indicated taxonomy species are relatively distant from *Pterostichus melas*, 12 peptides turned out to be neuropeptides belonging to known families including pyrokinin, FMRFamide, NPF, allatotropin, NP, antidiuretic factor, peptide hormone 1, and peptide tarsal-less AA (Appendix A). NeuroPIpred predictive tool was used to recognize the neuropeptides among those identified (https://webs.iiitd.edu.in/raghava/neuropipred/).

The search was performed using SVM threshold: 0.0 and Natural model. A score of 2.7 assigned by the Machine Learning Algorithm allow us to label 12 sequences as putative neuropeptides. Figure 1 shows MS/MS spectrum of pyrokinin-related peptide DGAETPGAAASLWFGPRV-Amide (*m/z* 1800.94). This signal dominates female chemical profiles in both control and treated samples; conversely, it isn’t detected in control as well treated male samples. Pyrokinins (*m/z* 1800) are a family of well-studied neuropeptides with myotropic, pheromonotropic, and melanotropic roles in several insects [36], involved in myo-stimulatory activity on visceral muscle [37], including oviduct muscle [38], in pheromone biosynthesis [39] and induction or inhibition of diapause hormone (DH) [40]. FMRFamide (*m/z* 1201.54) peptides are pleiotropic and thus participate in many processes including the stimulation of various muscles [41,42] and the promotion of the stress-induced sleep [43].

They regulate oogenesis and oviposition in the hemipteran *Rhodnius prolixus* [44] and heart physiology in Diptera [45]. In *Tenebrio molitor* and *Zophobas atratus*, they have myotropic effects in the regulation of contractile activity of the heart, ejaculatory duct, oviduct and the hindgut [46]. Allatotropins (*m/z* 1367.77) and allatostatins such as allatostatin-3 (ALL3_RHOPR, *m/z* 1846.97), identified in the haemolymph of *P. melas*, have inhibitory or stimulatory effects, respectively, on juvenile hormone biosynthesis in the corpora allata [47]. The role of these peptides has also been demonstrated in regulating several other functions including release of digestive enzymes in the midgut [48], myostimulation influencing the physiology of hearts in the dorsal vassel, gut, and reproductive system. The allatostatin effects have been considered on the egg movements in the oviducts and food transit in tenebrionid beetles [49]. Some neuro-derivate factors in insects are involved in the maintenance of homeostasis. There are numerous families of diuretic and anti–diuretic peptides involved in controlling active transport processes of the gut including the creation of urine in the Malpighian tubules and the uptake of water and salts in the hindgut [50]. Antidiuretic factor “ADFa peptide” (ADFA_TENMO, *m/z* 1542.80) has not been detected outside coleoptera; it is probably the fragment of larger proteins that are not normally liberated and part of regulatory functions [51]. Antimicrobial peptides (AMP) are small proteins having a broad range of activity against bacteria, fungi and viruses as an essential factor for innate immune response [52]. Bombolitin (*m/z* 1862.19), first isolated from *Apoidea* (Hymenoptera) venoms, is involved in the mast cell degranulation also showing as bactericide and fungicide properties [53]. Vespulakinin 1 (*m/z* 1960.11) described in *Vespula maculifroms* is a vasoactive peptide [54]. Phospholipases A2 (PLA2, *m/z* 2637.59) have been described in various hymenopteran species and act as neuro- and myotoxins by hydrolyzing membrane phospholipids of motor nerves and subsequently leading to cell lysis [55]. Metalnikowin IIB (*m/z* 2040.10) is a proline-rich antimicrobial peptide [56], first identified in *Palomena prasina* [57]. Cecropins (*m/z* 4083.40) first isolated from the haemolymph of pupae in giant silk moths, *Hyalophora cecropia*, were recognized as a part of immune response in Lepidoptera, Coleoptera, and Diptera [58]. Cecropins are released by fat bodies and haemocytes in haemolymph and have broad range of antimicrobial activity against bacteria and fungi [59]. Cupiennin-1c (*m/z* 3770.22) is a cytolytic peptide belonging to a family of antimicrobial peptides isolated from *Cupiennius salei*, displaying lytic activity towards bacteria, trypanosomes, plasmodia, human blood cells, and cancer cells [60]. Paralytic peptide (*m/z* 2476.16), first in lepidopteran species, belongs to the cytokine family. It delays larval growth, induces local paralysis around the wound reducing the loss of haemolymph, regulates the immune responses [61], and boosts cell proliferation and morphological variation [62]. Theraphotoxins including U3-theraphotoxin-Hhn1r (*m/z* 3671.70) and dermonecrotic toxins such as LgSicTox-beta-LOXN4 (*m/z* 4019.92) have been identified first in spiders to show highly selective inhibitor activity on cell ion channels and cellular lysis, respectively. Serine protease inhibitors (serpins, *m/z* 3750.72) are functionally distinct, and structurally conserved proteins present in all higher eukaryotes. They are involved in a wide array of physiological functions, including development, innate immune response, reproduction, and host-pathogen interactions [63]. Cell signaling pathways include small peptides such as peptide tarsal-less AA (*m/z* 3722.89) that coordinate multiple biological processes of cell division, survival, migration and differentiation, giving rise to the embryonic formation of the organs or regulating the innate immune responses [64]. The peptide tarsal-less AA has been studied in *Drosophyla* to be involved in leg morphogenesis [65], while in the silkworm *Bombix mori*, it participates in the development of the trachea, silk glands, and Malpighian tubules and is involved in the response to infections of pathogens [66].

### 2.2. Bioinformatic Analysis

Neuropeptides are responsible for the regulation of several processes, such as homeostasis, development, metabolism, and reproduction, and are produced from larger precursor proteins called prepropeptides [67,68]. Some of the neuropeptides detected in the haemolymph sample perform the duty for multiple functions arranging, as transmitters and co-transmitters with other neuropeptides, several biological process. Protein association network analysis for dynamic/transient interactions of components in signaling and regulatory pathways may be useful to evaluate the hexapoda’s physiological state. Notwithstanding, the large phylogenetic distance across these huge and various classes, neuropeptides display clear similarity among insect taxa [67,68]. In fact, the structure of the genes encoding both neuropeptides and receptors are highly conserved during evolutionary history [69].

Bioinformatic data-analysis is performed using ID of prepropeptide for the analogous reference species *Drosophila melanogaster*, which is the only other higher insect whose complete genome sequence is available in the public domain. Today at least 50 genes encoding precursors of neuropeptides, peptide hormones, and protein hormones are reported in *D. melanogaster*. Protein association network analysis for both male and female samples was performed by STRING database (https://string-db.org/). For female control samples, the network is composed of 17 nodes (prepropeptides) and 13 edges (interactions); expected number of edges 1 and PPI enrichment *p*-value: 3.38 × 10^−10^. To create this network, a value of 3 for the MCL clustering coefficient was chosen. Two protein clusters are observed, the main protein network showed the set composed by 6 nodes (AstA, Dh44, FMRFa, Hug, Ilp1, sNPF) (Figure 2A) is referred to “Neuropeptide signaling pathway, and insulin-like superfamily”. Accordingly, two networks are observed for male (Figure 2B), and the principal network is composed by 5 nodes (AstA, Dh44, FMRFa, Ilp1, sNPF). The central node in the networks related to both male and female is the neuropeptide sNPF (Figure 2A,B). This feature can’t be explained attributing a unifying global function to sNPF, but underlies that this neuropeptide accomplishes several distinct roles, and it has distributed functions. In fact, Neuropeptide F (sNPF) (*m/z* 1213) has been suggested to have multiple functional roles in foraging, feeding, alcohol sensitivity, stress, aggression, reproduction, learning function, and circadian rhythm [70,71]. The NPF role in male fertility and female oocyte maturation is agree with the specimens analyzed that were in the reproductive phase of their life cycle.

### 2.3. Chemical Component Profile of Control and PND-Treated Groups

The chemical component profile of haemolymph specimens (Figure 3) is affected by the individual variability (sex, hormonal variation, and response to treatment). The visual comparative analysis of the spectra, over time after PND exposition, shows changes of the overall profile with consistent modifications in specific regions. Male profiles appear more complex and the feature seems to be related to the time and the treatment. The change overtime of control male profiles is probably related to the insect growth. Otherwise, female profiles (control and treated samples) all seem similar, less complex and dominated by the signal of pyrokinin (*m/z* 1800). This peptide is absent in male profiles (Figure 3). The detection of pyrokinin in haemolymph of the females may likely related to its myotropic action on the ovipositor contraction during egg laying, as observed in cricket *Gryllus bimaculatus* [72].

To analyze the relationship between spectra and PND treatment, a pre-processing step is adopted. Nine replicates are aligned and average, in order to obtain a unique representative profile for each specimen used in subsequent chemometric analysis. The data matrix, obtained using the signals of the *m/z* values selected on the basis of a threshold greater than 400 in terms of signal-to-noise in at least four of the analyzed samples (43 variables), was subjected to principal component analysis (PCA). PCA is a chemometric tool widely used in data exploratory analysis in order to have an overview of data.

PCA transforms the original variables, using an orthogonal linear transformation, to a new set of uncorrelated variables known as principal components (PCs). Representation of the principal component scores and loadings in a bidimensional plot can be used as powerful visualization tool, pointing out patterns hidden in the data set and finding possible correlations between variables. The scores of the samples and the loadings of the variables on the two first principal components are plotted (Figure 4). The information retained is 39.39% of the total variance. The plot of scores obtained by PCA highlights that the female control samples are very similar to each other, while the male control ones are placed in points far from each other, especially for the control sample of the treatment at 7 days (Figure 4A). In any case, all the male control samples are different from female control samples mostly along the first principal components. The difference between these samples can be attributed to the variables placed on the far left and the far right of the loading plot, corresponding to compounds present at higher concentration in male and female samples, respectively (Figure 4B). In particular, the female samples have higher concentrations of peptide tarsal-less AA (*m/z* 3722.89, Table 1) and pyrokinin (*m/z* 1800.94). On the other hand, higher concentrations of phospholipase A1 verutoxin-1 (Fragment) (*m/z* 2637.59), vespulakinin-1 (*m/z* 1960.11), serine protease inhibitor 2 (*m/z* 3750.72), bombolitin-3 (*m/z* 1862.19), M-lycotoxin-Ls4a (*m/z* 2211.28), dermonecrotic toxin LgSicTox-beta-LOXN4 (*m/z* 4019.92), and peptide hormone 1 (*m/z* 2010.94) were found in the male samples (Table 1, Figure 4A,B).

The comparison between the females from the control group and treated samples underlines that there is no significant difference between samples belonging to the group exposed at 7 days and 21 days, whereas a shifting can be observed along PC1 going from control sample to treated one at 2 days from the initial exposure, which indicates an increase of the signal for the *m/z* values on the left of the plot of loadings. Therefore, the exposure at 2 days involves the concentration increase of phospholipase A1 verutoxin-1 (Fragment) (*m/z* 2637.59), vespulakinin-1 (*m/z* 1960.11), serine protease inhibitor 2 (*m/z* 3750.72), bombolitin-3 (*m/z* 1862.19), M-lycotoxin-Ls4a (*m/z* 2211.28), dermonecrotic toxin LgSicTox-beta-LOXN4 (*m/z* 4019.92), and Peptide hormone 1 (*m/z* 2010.94), as well as of the unknown compounds placed on the left of the plot. The greatest difference is observed in males at 7 days of exposure compared to the control ones. In particular, the two samples have very different values for both first and second principal component demonstrating that the treatment involves a decrease of the signal for the variables on the left of the plot of loadings (high negative loading values on PC1) and, at the same time, an increase for the variables at the bottom of the loading plot (high negative loading values on PC2, M-lycotoxin-Ls4a of *m/z* 2211.28, and Cecropin of *m/z* 4083.40, among the identified compounds). The samples after 21 days from the initial exposure showed a shifting similar but in opposite direction to that observed for the female samples at 2 days of exposure. Finally, a different behavior is detected for the samples belonging to the treatment at 2 days, for which a clear separation occurs exclusively on the second principal component. In this case, the position of the treated sample at higher score values on PC2 means that the treatment determines a decrease of the signal for the *m/z* values being at the bottom of the loading plot, among which there are the aforementioned peptides M-lycotoxin-Ls4a (*m/z* 2211.28), and Cecropin (*m/z* 4083.40).

In order to better evaluate the effect of treatment on the fingerprint profiles obtained by mass spectrometry analysis, another data matrix was constructed using the ratio of the signals of the PND-treated specimens to the signals of the control specimens for each selected *m/z* value. The application of PCA to this matrix allows to obtain a score plot (Figure 5) in which the variation of the MS profile respect to PND-treatment is clearly shown. Treatment certainly involves different consequences for males and females. In fact, for females the most important variation is substantially along the first principal component, while for males the variations concern only PC2. Therefore, the variables responsible for the modifications of profiles due to treatment are different for males and females. However, a trend common to males and females can be observed. In fact, for both types of samples the greatest variation is noticed passing from the first to the second treatment, while the third treatment are between the first two. This indicates that the involved variables first are subjected to a great variation and then return to assume intermediate values. In particular, for females, a significant decrease of signal ratio passing from the first to the second treatment is observed for the variables placed on the far right of the loading plot shown in Figure 5 (1217.18; 1800.94; 1822.60; 1862.19; 1871.69; 1960.11; 2040.10; 2288.06; 2476.16; 2637.59; 3588.24; 3685.98; 4019.92). On the other hand, for males the variables that show a substantial decrease are the following: 3650.00, 4026.30, 4078.35, and 4233.57. A total of 13 and 4 differentially regulated metabolites were found in female and male, respectively. There are interesting dysregulation patterns in neuropeptides with significant changes observed resulting from PND exposure. However, for males, the treatment at 21 days involves also the increase of signal values for the variables at the bottom of the loadings plot (high negative loading values on PC2), that is 3649.04, 3611.07, 4133.56, and 2822.56. PCA elaboration highlighted differences in the responses of *P. melas* according to gender.

### 2.4. Probing Peptide Changes after PND Treatment

Figure 6 A,B depicts the significant expression changes of the identified peptides in haemolymph from females and males of *P. melas* as a ratio to the control after each exposure duration. These ratios are expressed as a log2 ratio for easier viewing (positive results show upregulation, negative show downregulation). There are interesting and significant peptide changes resulting from PND treatment. There are also interesting general trends observed between the two groups. Some peptides show a constant pattern of up-downregulation over 21 days, others show a hyperarousal pattern. In these cases, the peptide is increased (Metalnikowin-2B, VDKPDYRPRPWPRNMI, and M-lycotoxin-Ls4a, IASHLAFEKLSKLGSKHTML, Figure 6A) at one time point and returns to baseline. A significant downregulation of Paralytic peptide 2 (ENFAGGCTPGYQRTADGRCKPTF, Figure 6B) is observed in male sample after 2 days of exposure suggesting that PND is able to suppress the PP-dependent induction analogously to a pharmacologic p38 MAPK inhibitor [61]. In female samples the downregulation of Paralytic peptide 2 is observed only after 21 days of treatment. In particular, in females of *P. melas* is observed the upregulation of PP2 in first two days, followed by an important decrease leading to significative downregulation after 21 days of treatment. The temporal shift in upregulation between the groups indicate that distinct pathways are involved in their regulation. The observed trend can be explained considering that PP2 participates to the physiological processes, such as embryonic morphogenesis and larval growth rates. In particular, the observed upregulation of pyrokin and PP2 neuropeptides after 2 days of exposure suggests the activation of signaling pathways to increase muscle contractions in the ovipositor activity and embryonic morphogenesis (Figure 6, panel A).

The downregulation of PSK (Peptide hormone 1, SDLTWTYQSPGDPTNSKN) in males, that become significant only after 21 days, is probably correlated to captivity condition and food copiousness. In fact, the gut peptide PSK acts antagonistically to the hunger signal provided by the adipokinetic hormone (AKH) inhibiting the pTRPγ channel, which is activated under conditions of food shortage [73].

In response to PDN exposure, significant changes occur in males, resulting in downregulation of serpin (EISCEPGTTFQDKCNTCRCGKDGKSAAGCTLKACPQ, *m/z* 3750.72) at the time points 2 and 7 days, turning to baseline after 21 days. In females, serpin is in baseline at the time points 2 and 7 days, but significantly upregulated after 21 days. Serpins carry out various physiological functions in insects, including development, digestion, host-pathogen interactions, and innate immune response. Serpin plays a key role in immune pathways, its involvement in the regulation of the Toll pathway and prophenoloxidase activation is well documented [74]. Its downregulation was observed in *Plutella xylostella* after fungal and viral infection, and treatment with mycotoxin [74].

The effect of PND exposure is less important in females than in males, suggesting than females are less sensitive. The observed differences in dysregulation of peptides between males and females highlight that they probably activate different mechanisms of defense for uptake or elimination of toxicants. These data agree with the general mechanism of uptake and elimination of toxicants reported for Arthropods [75]. The females lower their physiological exposure, minimizing any toxic effects of PND in the tissues, by depositing toxicants into their eggs. In males, which do not have this physiological advantage, a significant downregulation of related to immune and defense peptides (M-lycotoxin-Ls4a, Peptide hormone 1, Paralytic peptide 2, and Serine protease inhibitor 2) is observed.

## 3. Materials and Methods

### 3.1. Sample Collection and Treatment

Adults of *P. melas* were collected in an organic olive grove (39°59′27.56′′ N, 16°15′32.64′′ E, 1202 m a.s.l. San Marco Argentano, Calabria, Southern Italy) in October 2019 by using in vivo pitfall traps (plastic jars 9 cm in diameter) containing fruit as an attractant. In the laboratory, beetles were identified by using a dichotomous key, separated by sex and kept in 5 L plastic boxes that were filled to a depth of 6 cm with soil from the capture site. Males and females were separately housed and maintained to be acclimatized for 1 month at 60% relative humidity (rh), under a natural photoperiod, and at room temperature. They were fed mealworms and fruit (organic apples) ad libitum.

A commercial formulation of pendimethalin (PND; Activus EC, product n° HRB00858-39; active ingredient 330 g/L^−1^) was tested in males and females exposed for 21 days at a recommended field rate (4 L per ha, for cereal and vegetable crops), considering that the half-life of pendimethalin ranges from 24.4 to 34.4 days in acidic sandy soil [1,4]. The experimental design included 6 × 2 plastic boxes (180.5 cm^2^) filled with the clean sandy soil (pH 5 approximately) from the capture site. Exposure was carried out by spraying the PND solution (7.2 µL of Activus in 14 mL of distilled water) with a pipette on the soil surface of 6 boxes for the treated groups (3 boxes for male and 3 boxes for female, each box containing 10 insects) to simulate the field exposure by contact with the contaminated soil. The control groups, comprising 6 boxes of (3 boxes for male and 3 boxes for female, each box contain 10 insects), were sprayed with distilled water. Males and females, kept separately, were introduced 15 min after the PND solution (or water) was sprayed. Thereafter, 4 boxes (2 control and 2 treated) were used after to 2, 7, and 21 days, respectively. A single application of PND-based herbicide was carried out at 0 day, for treated groups.

### 3.2. Haemolymph Collection

Males and females from both control (*n* = 9) and treated (*n* = 9) groups were randomly chosen at 2, 7 and 21 days after the initial exposure and anaesthetized in a cold chamber at 0 °C for 3 min. Haemolymph was collected by puncturing cold anaesthetized beetles ventrally at the pro-mesothorax articulation with a 29-gauge needle. A pool of 15 μL of haemolymph was collected from three specimens, promptly diluted 1:10 into cold ammonium bicarbonate saline solution (NH_4_HCO_3_, 50 mM; Sigma-Aldrich, Darmstadt, Germania) and centrifuged at 1700 rpm for 10 min at 4 °C. Cell-free haemolymph obtained as supernatant was collected and stored at −20 °C before chemical analyses.

### 3.3. Sample Preparation

An aliquot of each sample (10 µL) was added with 90 µL of solution CHCl_3_/CH_3_OH diluted 1:3 (*v*/*v*), yielding a lipophilic supernatant fraction and a pellet. The supernatant fractions were analyzed by mass spectrometry. A total of 10 µL of lipophilic supernatant fractions were completely air-dried at room temperature and solubilized in 10 μL of α-cyano-4-hydroxycinnamic acid as MALDI matrix (α-CHCA, 5 mg/mL; H_2_O/CH_3_CN, 50:50, v:v; 0.3% TFA). Each sample was spotted onto a 384-well insert Opt-TOF TM stainless steel plate (AB SCIEX, Darmstadt, Germany) and a rapid drying protocol [76,77] was adopted to reduce the inhomogeneous co-crystallization of the analyte with the matrix.

### 3.4. MALDI MS Analysis

Mass spectrometry analyses (MS) were performed by 5800 MALDI TOF-TOF Analyzer (AB SCIEX, Darmstadt, Germany) equipped with an Nd: YLF Laser with λ = 345 nm wavelength of <500 ps pulse length and to 1000 Hz repetition rate. MS analysis was performed in positive reflectron mode, mass accuracy of 10 ppm and a mass range 900–5000 Da were set to record the untargeted fingerprint profile of low molecular weight species. The peptide mass standards kit (Calibration Mixture 1, AB SCIEX) was used to calibrate the MALDI TOF/TOF mass spectrometer. Spectra with signal-to-noise below 200 were automatically discarded by the instrument and at least 4500 laser shots were typically accumulated with a 400 Hz laser pulse. Each haemolymph sample was spotted three times and three spectra were acquired for each spot in order to obtain a set of nine data point (9 row data). Each spectrum results from the accumulation of 4500 laser shots and the laser was set to continuous random movement to obtain a uniform crystal ablation. The corresponding set of nine data points (9 spectra) was merged and averaged to obtain a unique representative profile for each specimen. All spectra were processed using Data Explorer version 4.11 (AB Sciex) and all nine data points were merged and averaged to obtain a data matrix useful for statistical analysis. A comparative quantification analysis was performed. Collision-induced dissociation (CID)-MS/MS analysis were performed to characterize the selected ion species. MS/MS spectra were performed at a collision energy of 1 kV, and ambient air was used as the collision gas (10−6 Torr) acquiring up to 6000 laser shots e averaging a pulse rate of 1000 Hz.

### 3.5. Database Proteomics and Targeting Predictions

The MS/MS data were processed to produce pick lists for searching in the SwissProt database using the MASCOT search software (© Matrix Science 2021, http://www.matrixscience.com). The mass tolerance of the parent and fragments for MS/MS data search was set at 20 ppm and 0.20 Da, respectively. During database search the following query were considered “Metazoa (Animals)” taxonomy and choosing “noCleave”. A peaklist of 50 ions of 10% higher intensity than the noise level was generically used for database searching. Methionine oxidation was included in the variable modifications. Although several MS/MS spectra showed intense and well-resolved ion signals, all spectra were manually checked to validate MASCOT results. Neuropeptide homologs are determined by Domain Enhanced Lookup Time Accelerated Basic Local Alignment Search Tool (DELTA-BLAST), limiting the searches to homologs in Drosophila melanogaster. Network analysis was performed by STRING (Search Tool for the Retrieval of Interacting Genes) software (v. 11) (http://stringdb.org/). NeuroPIpred was used to evaluate potential neuropeptides in *P*. *melas* (https://webs.iiitd.edu.in/raghava/neuropipred/).

### 3.6. Statistical Analyses

The MS based untargeted strategy gives an impartial approach on every peak recorded and assures a comprehensive collection of data, without selection criteria. To evaluate possible differences among males and females from both control and PND-treated groups, principal component analysis (PCA) was performed by Statistica 8.0 software package (StatSoft 2007 Edition, Tulsa, OK, USA). The use of PCA allows investigating complex samples and novel metabolic pathways [78]. The analysis included exact mass and IMS clustering. To ensure the quality of the results after data processing, only the signals present in all three biological replicates were considered for differential abundance analysis.

Peptides with significant Student’s *t*-test (*p*-value < 0.05, two-tailed) results were considered differentially expressed. Peptides were considered upregulated if the Log2 was greater than 0.5 and downregulated if the Log2 was lower than −0.5.

## 4. Conclusions

Exposure of *P. melas* to PND based herbicide altered the metabolic profile of haemolymph. This result indicates that PND interferes with the epigenetic mechanisms of *P. melas*, and may sabotage the immune system and slow down its development. PCA elaboration highlighted important differences between males and females’ responses after exposure to PND indicating evidence for a sex-specific detoxification response. The metabolomic response highlights the effects of herbicides on non-target animal species, suggesting that their ecological role might be compromised.

## Figures and Tables

**Figure 1 molecules-27-04645-f001:**
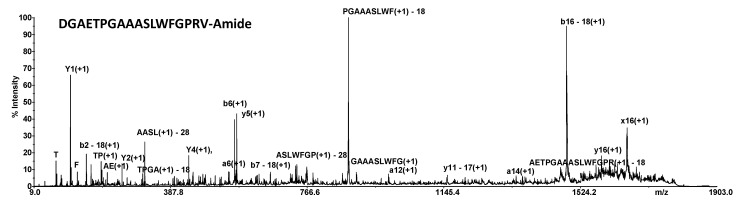
MALDI MS/MS of the *m/z* 1800.94.

**Figure 2 molecules-27-04645-f002:**
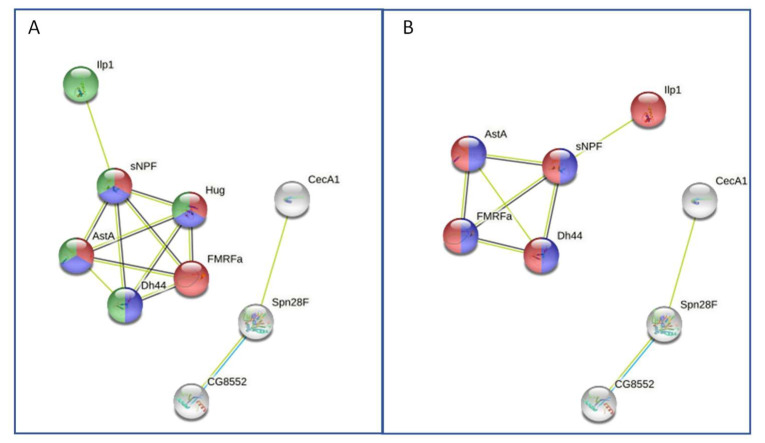
Protein–protein interaction network by STRING software version 11, https://string-db.org/ (panel (**A**), female; panel (**B**), male).

**Figure 3 molecules-27-04645-f003:**
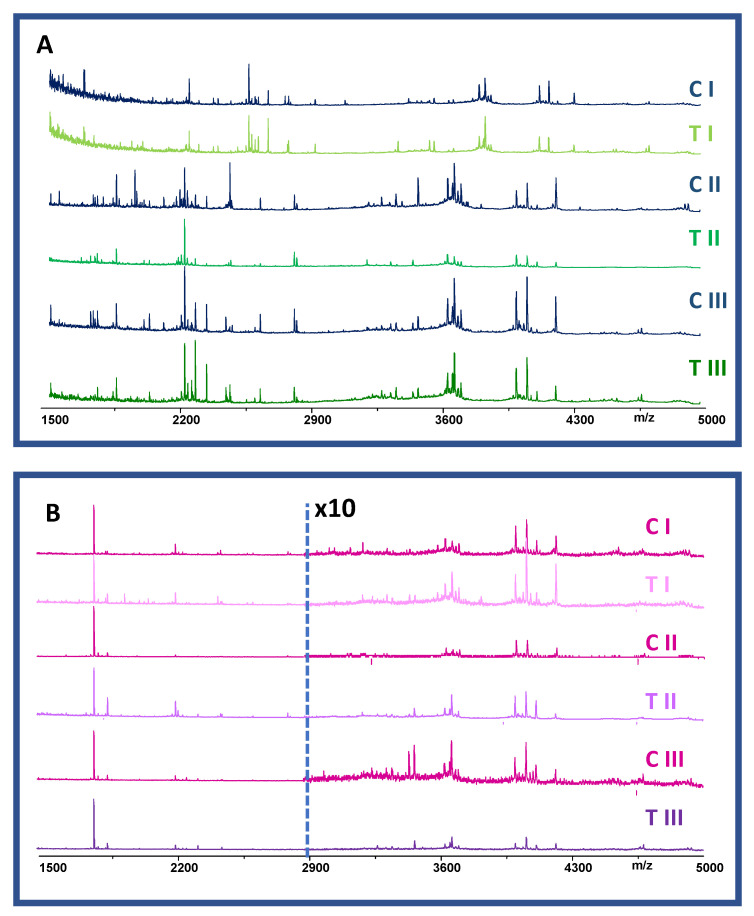
Untargeted molecular profiles of control (C) and treated (T) male insects (panel (**A**)) and female insects (panel (**B**)), after 2 (I), 7 (II), and 21 (III) days of PND treatment.

**Figure 4 molecules-27-04645-f004:**
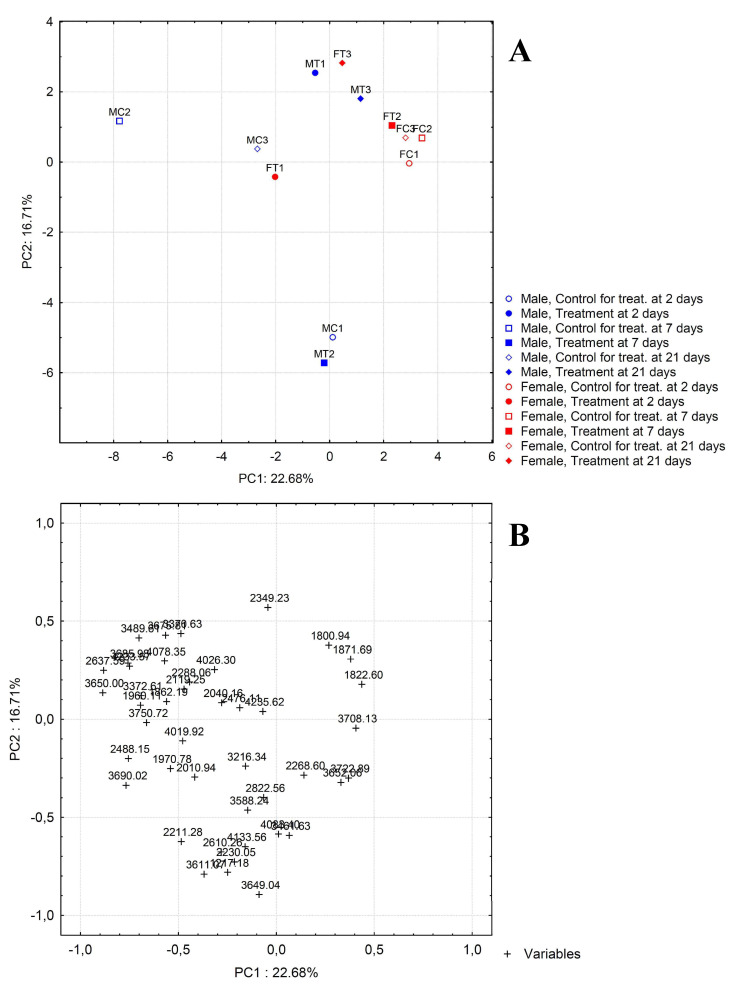
Scores (**A**) and loadings (**B**) plots obtained by Principal Component Analysis applied to mass spectrometry dataset.

**Figure 5 molecules-27-04645-f005:**
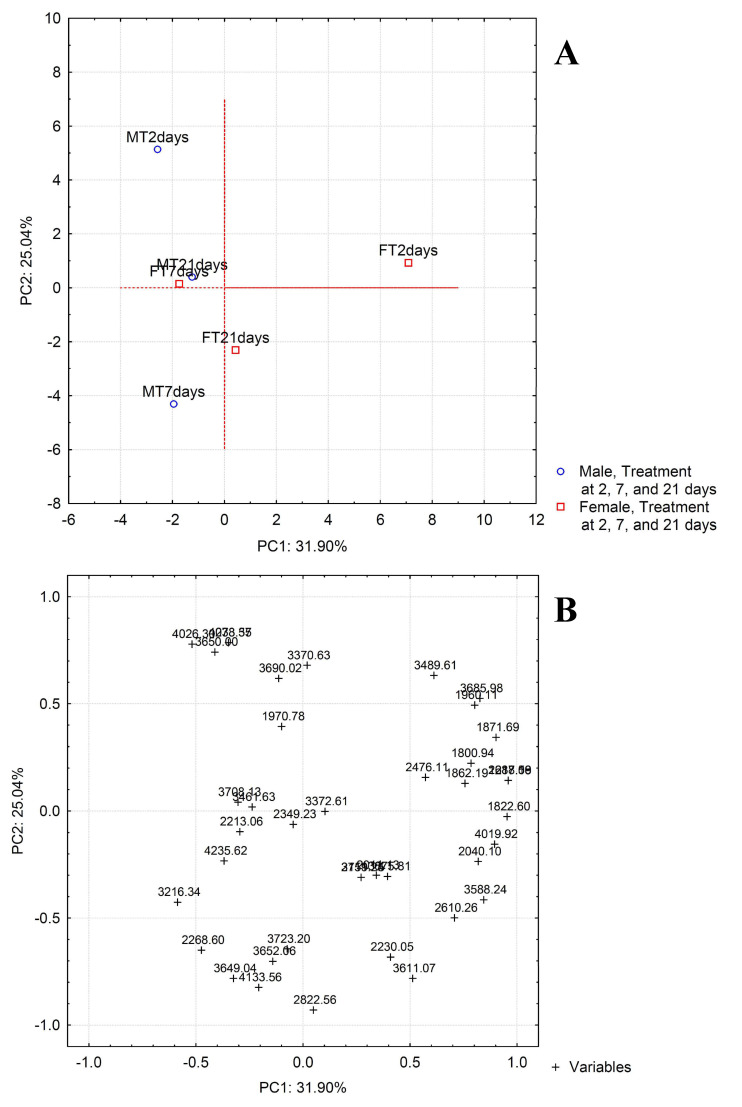
Scores (**A**) and loadings (**B**) plots obtained by Principal Component Analysis applied to the ratio of the signals of treated specimens to the signals of the control specimens.

**Figure 6 molecules-27-04645-f006:**
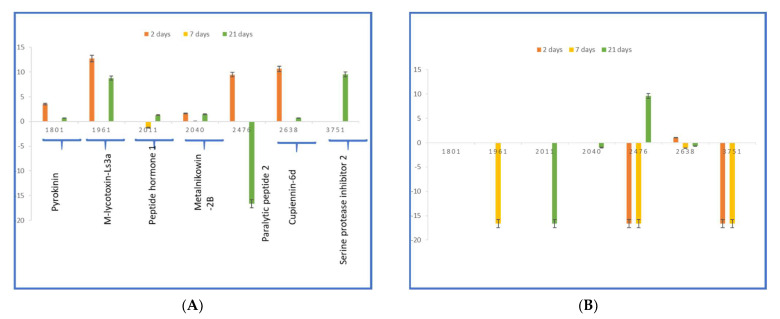
The ratio for the intensity after 2, 7, and 21 days of PND exposure relative to the control is depicted on the *y*-axis as a log2 ratio to clearly see up/downregulation of peptides relative to the *x*-axis. Panel (**A**,**B**) show up/downregulation of peptides for female and male groups, respectively. The error bars denote the standard error of the mean (*n* = 3).

**Table 1 molecules-27-04645-t001:** Identified proteins by MALDI MS/MS and Mascot software by Matrix Science (www.matrixscience.com). The MS/MS data were processed using a mass tolerance of 20 ppm and 0.2 Da for the precursor and fragment ions, respectively. **^a^** According to “UniProtKB” (http://www.uniprot.org/).

	ID^a^	Name ^a^	Sequence	Exact	*m/z*
1.	FAR14_SARBU	FMRFamide-14	DPHHDFMRF	1201.5213	1201.54
2.	NPF1_LEPDE	Neuropeptide NPF-1	ARGPQLRLRF	1213.7282	1213.75
3.	ALLTR_ACRHI	Allatotropin-related peptide	GFKNVALSTARGF	1367.7435	1367.77
	ALLTR_BANDI	Allatotropin-related peptide	GFKNVALSTARGF		
	ALLTR_EUSSE	Allatotropin-related peptide	GFKNVALSTARGF		
	ALLTR_NEZVI	Allatotropin-related peptide	GFKNVALSTARGF		
	ALLTR_ONCFA	Allatotropin-related peptide	GFKNVALSTARGF		
	ALLTR_PENRU	Allatotropin-related peptide	GFKNVALSTARGF		
	ALLTR_PERAM	Allatotropin-related peptide	GFKNVALSTARGF		
	ALLTR_PYRAP	Allatotropin-related peptide	GFKNVALSTARGF		
4.	ADFA_TENMO	Antidiuretic factor	VVNTPGHAVSYHVY	1542.7705	1542.80
5.	TXS6D_CUPSA	Short cationic peptide-6d	INKYREWKNKKN	1620.8974	1620.92
6.	PPK_SCHGR	Pyrokinin	DGAETPGAAASLWFGPRV—Amide	1800.9032	1800.94
7.	ALL3_RHOPR	Allatostatin-3	QVSLKYPEGKMYSFGL	1846.9413	1846.97
8.	BOL3_BOMPE	Bombolitin-3	IKIMDILAKLGKVLAHV	1862.1665	1862.19
9.	BRK_VESMC	Vespulakinin-1	TATTRRRGRPPGFSPFR	1960.0741	1960.11
10.	LYC1_LYCSI	M-lycotoxin-Ls3a	GKLQAFLAKMKEIAAQTL	1961.1257	1961.16
11.	PH1_PERAM	Peptide hormone 1	SDLTWTYQSPGDPTNSKN	2010.9045	2010.94
12.	MK2B_PALPR	Metalnikowin-2B	VDKPDYRPRPWPRNMI	2040.0601	2040.10
13.	LYC40_LYCSI	M-lycotoxin-Ls4a	IASHLAFEKLSKLGSKHTML	2211.2323	2211.28
14.	PAP2_SPOEX	Paralytic peptide 2	ENFAGGCTPGYQRTADGRCKPTF	2476.1138	2476.16
15.	PA11_VESVE	Phospholipase A1 verutoxin-1 (Fragment)	GLLPKVKLVPEQISFILSTRENR	2637.5455	2637.59
16.	TXC6D_CUPSA	Cupiennin-6d	FINTIKLLIEKYREWKNKQSS	2638.4721	2638.52
17.	HN423_CYRHA	U3-theraphotoxin-Hhn1r	DCAGYMRECKEKLCCSGYVCSSRWKWCVLPAP	3671.6222	3671.70
18.	MSPI2_MELSA	Serine protease inhibitor 2	EISCEPGTTFQDKCNTCRCGKDGKSAAGCTLKACPQ	3750.6476	3750.72
19.	TXC1C_CUPSA	Cupiennin-1c	GFGSLFKFLAKKVAKTVAKQAAKQGAKYIANKQTE	3770.1484	3770.22
20.	TALAA_DROME	Peptide tarsal-less AA	LDPTGTYRRPRDTQDSRQKRRQDCLDPTGQY	3722.8169	3722.89
21.	BX4_LOXGA	Dermonecrotic toxin LgSicTox-beta-LOXN4	ADSRKPDDRYDMSGNDALGDVKLATYEDNPWETFK	4019.8357	4019.92
22.	CEC_CALVI	Cecropin	GWLKKIGKKIGRVGQHTRDATIQGLAVAQQAANVAATAR	4083.3167	4083.40
23.	DIUH1_TENMO	Diuretic hormone 1	SPTISITAPIDVLRKTWEQERARKQMVKNREFLNSLN	4369.3566	4369.44

## Data Availability

Not applicable.

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
