# Peer review of "Mass Spectrometry-Based Peptide Profiling of Haemolymph from Pterostichus melas Exposed to Pendimethalin Herbicide"

_molecules, 2022, doi:10.3390/molecules27144645_

Round 1

Reviewer 1 Report

The current manuscript describes non-targeted analysis of the proteomics of a carabid beadle species as affected by the herbicide Pendimethalin, PND.  The study uses MALDI-MS, itself not a common method and not the preferable for bottom up proteomics analysis.  By LC-MS/MS, peptides with similar masses can be separated by a reverse phase chromatography, the results of which are more informative since they include m/z, retention time and MS/MS information.  Nevertheless for the scope of this journal the experiments are worthy of study since the authors compare the effects of PND at three different time points and by sex.  Remarkably, the effects are more pronounced in male insects than in female ones.  My concerns which I hope get addressed, are outlined below:

1. the authors could have use LC-MS/MS.  The analysis might require more sample since the peptides are separated during chromatography which would produce weaker signals, it might be the authors do not have access to a uHPLC system that can be interfaced to the mass spectrometer, but this is something to consider.

2. I am not an expert in chemometric analysis but have done so for other metabolomics projects.  The current standard for non-targeted analysis is Metaboanalyst.ca, a very extensive program that can do much better visualization and analysis at zero cost, and which requires no special formatting of the data and hardly any training.  Did you authors Metaboanalyst?

3. The data in Figure 3B for the female samples appear to be identical across the samples, thit might be due to the strong peak at low mass, so rescaling is necessary.  Both panels in this figure are missing the m/z axis units.

4. The interpretation of the MS data required searching the data against the Mascot DB.  But it could be that peptides which not available in the DNA database for this beetle and could show significant deviations between control and treatment.  Were the neuropeptides part of the search list as peptides and was that observed in the present study?

5.  The choice of peak ratios for PCA for both Males and Females is significant in highlighting the differential effects between the two sexes caused by the PND treatment.  How many principal components were necessary to analyze the data in Figures 4 and 5?  The presented data includes only up to 40% of the statistical effects.  Perhaps the rest of the statistical dump can be added as Supplementary material.

6. I am confused whether a second PND spray treatments were carried out for 7 and 21 days to both male and female beetles, or only a single treatment at said day (24 hrs) for both sexes, or maybe both experiments were done.  It is not clear from the discussion though the material section appears to indicate continuous treatments.

7. The hormone profiles for male and female beetles are quite different, regardless of the conditions of the samples. It might be useful to include in table 1 the ratios M/F or F/M for the neuropeptides hits.

8. The peptide sNPF plays a crucial role in the overall network analysis, yet it was not one of the MS/MS targets shown in table 1.  I am not clear if a relation would be expected here; in lines 212-214 the statement does not appear to follow.

9.  Figures 4 and 5 present the results of the raw and normalized PCA 2 major components.  The quality of these graphs can be greatly improved: for example the PCA plot has 6 data points (?) but only three can be distinguished.  The captions need to be moved into the description of the figures.

10. Figure 6 is lacking the scale of the y-axis. There are several peaks that point to a problem with the observation of the peaks.  For example, m/z-1961.16 becomes elevated after 2 days, then it levels off at 7 days and then it reappears at 21 days, in the female samples.  This is very hard to explain and might suggest that at 7 days the peak is there but for some experimental or technical reason it was not observed.  Likewise in the male samples the same peak is unchanged after 2 days (or not observed!), only to show a significant decrease after 7 days only to reappear at 21 days (or not observed).  The latter case seems feasible but one should expect to see bars for the three time points in all the samples (except for 1800.94 which is not present in males).  Could the reason be that they authors use a strict log 2 > 0.5 criterion or that the peaks were not reproducible in the replicates?

11.  The higher resistance of females to response could involve a less efficient mechanism of metabolism/excretion in males compared to females.  It would have been appropriate to measure levels of the herbicide along with the lipophilic peptides for the samples in this study.

12.  The authors speculate epigenetic effects of exposure of the beetles to PND.  Is this the case because of the known downstream action of the identified neuropeptides or is this just a general case of effects not involving genome mutations?

Thanks

Author Response

Reviewer: 1

Comments and Suggestions for Authors

The current manuscript describes non-targeted analysis of the proteomics of a carabid beadle species as affected by the herbicide Pendimethalin, PND. The study uses MALDI-MS, itself not a common method and not the preferable for bottom up proteomics analysis. By LC-MS/MS, peptides with similar masses can be separated by a reverse phase chromatography, the results of which are more informative since they include m/z, retention time and MS/MS information. Nevertheless, for the scope of this journal the experiments are worthy of study since the authors compare the effects of PND at three different time points and by sex. Remarkably, the effects are more pronounced in male insects than in female ones. My concerns which I hope get addressed, are outlined below:

- 1. the authors could have use LC-MS/MS. The analysis might require more sample since the peptides are separated during chromatography which would produce weaker signals, it might be the authors do not have access to a uHPLC system that can be interfaced to the mass spectrometer, but this is something to consider.

- 1 Authors thank Reviewer 1 for precious suggestion and constructive remarks. Nevertheless, the aim of the study was the evaluation of the chemical component profile of haemolymph from P. Melas exposed to PND. The experimental design was planned considering the advantages of high sensitivity for peptide analysis associated with MALDI TOF/TOF platform and comprehensive fragmentation information provided by high-energy collision-induced dissociation (CID). It is well-known that MALDI-TOF/TOF produces singly charged ions, accurate mass matching of the full-length, intact peptide, and allows confidently to identify peptides.

-2. I am not an expert in chemometric analysis but have done so for other metabolomics projects. The current standard for non-targeted analysis is Metaboanalyst.ca, a very extensive program that can do much better visualization and analysis at zero cost, and which requires no special formatting of the data and hardly any training. Did you authors Metaboanalyst?

 2 Authors thank Reviewer 1 for suggestion. AB SCIEX Data Explorer format is not supported by the free available MetaboAnalyst 5.1 (mzML, mzXML, CDF or mzData format in centroid mode), therefore the dataset was processed by Statistica 8.0 software package that is widely used in many fields.

- 3. The data in Figure 3B for the female samples appear to be identical across the samples, that might be due to the strong peak at low mass, so rescaling is necessary. Both panels in this figure are missing the m/z axis units.

-3 Figure 3B was modified as suggested.

  1. The interpretation of the MS data required searching the data against the Mascot DB. But it could be that peptides which not available in the DNA database for this beetle and could show significant deviations between control and treatment. Were the neuropeptides part of the search list as peptides and was that observed in the present study?

  1. The choice of untargeted strategy is aimed by the idea to obtain an impartial approach on every peak that is recorded during the MS analysis. This approach assures a comprehensive collection of data and, if it is combined with a multivariate statistical data analysis, allows the study and comprehension of highly complex samples. The advantage of an untargeted analysis is to obtain an unbiased means to evaluate the relationship between different molecules. It allows to achieve a comprehensive idea of a biological system and provides a starting point to assess the relationship between interconnected multiple pathways. Even if the species is not available into database, the strategy does not allowed deviations between control and treatment.

  1. The choice of peak ratios for PCA for both Males and Females is significant in highlighting the differential effects between the two sexes caused by the PND treatment. How many principal components were necessary to analyze the data in Figures 4 and 5? The presented data includes only up to 40% of the statistical effects. Perhaps the rest of the statistical dump can be added as Supplementary material.

  1. In general, a number of principal components equal to the number of original variables were obtained by PCA. However, the amount of information (variance) decreases as the order of the principal components increases. Therefore, the high-order PCs have low information content. Generally, the useful information is concentrated on the first (1-5) components. By considering the first two PCs we observe the maximum amount of information available on a plane in the multidimensional space of the original information.

  1. I am confused whether a second PND spray treatments were carried out for 7 and 21 days to both male and female beetles, or only a single treatment at said day (24 hrs) for both sexes, or maybe both experiments were done. It is not clear from the discussion though the material section appears to indicate continuous treatments.

-6. P 11, L 386-392, The sentences “The experimental design …………………………… was sprayed.” were modified as follow

“The experimental design included 6 x 2 plastic boxes (180.5 cm2) filled with the clean sandy soil (pH 5 approximately) from the capture site. Exposure was carried out by spraying the PND solution (7.2 µL of Activus in 14 mL of distilled water) with a pipette on the soil surface of 6 boxes for the treated groups (3 boxes for male and 3 boxes for female, each box contain 10 insects) to simulate the field exposure by contact with the contaminated soil. The control groups, comprising 6 boxes of (3 boxes for male and 3 boxes for female, each box contain 10 insects), were sprayed with distilled water. Males and females, kept separately, were introduced 15 min after the PND solution (or water) was sprayed. Thereafter, 4 boxes (2 control and 2 treated) were used after to 2, 7, and 21 days, respectively. A single application of PND-based herbicide was carried out at 0 day, for treated groups.”

  1. The hormone profiles for male and female beetles are quite different, regardless of the conditions of the samples. It might be useful to include in table 1 the ratios M/F or F/M for the neuropeptide’s hits.

  1. The hormone profiles for male and female beetles is well described in PCA analysis (P 8, L 248-262). Authors prefer don't add the ratios in table 1 (M/F or F/M) since it could be confusing for the reader.

  1. The peptide sNPF plays a crucial role in the overall network analysis, yet it was not one of the MS/MS targets shown in table 1. I am not clear if a relation would be expected here; in lines 212-214 the statement does not appear to follow.

  1. The peptide sNPF is listed in Table 1, row 2. P 6, L212-214 The sentence “By considering …….underlined.” was modified as follow

“The NPF role in male fertility and female oocyte maturation is agree with the specimens analyzed that were in the reproductive phase of their life cycle.”

  1. Figures 4 and 5 present the results of the raw and normalized PCA 2 major components. The quality of these graphs can be greatly improved: for example the PCA plot has 6 data points (?) but only three can be distinguished. The captions need to be moved into the description of the figures.

  1. High quality figures were uploaded in the revision procedure.

  1. Figure 6 is lacking the scale of the y-axis. There are several peaks that point to a problem with the observation of the peaks. For example, m/z-1961.16 becomes elevated after 2 days, then it levels off at 7 days and then it reappears at 21 days, in the female samples. This is very hard to explain and might suggest that at 7 days the peak is there but for some experimental or technical reason it was not observed. Likewise in the male samples the same peak is unchanged after 2 days (or not observed!), only to show a significant decrease after 7 days only to reappear at 21 days (or not observed). The latter case seems feasible but one should expect to see bars for the three time points in all the samples (except for 1800.94 which is not present in males). Could the reason be that they authors use a strict log 2 > 0.5 criterion or that the peaks were not reproducible in the replicates?

  1. Figure 6 was modified as suggested. Authors used the strict log 2 > 0.5 cutoffs in order to retain only high-quality features.

  1. The higher resistance of females to response could involve a less efficient mechanism of metabolism/excretion in males compared to females. It would have been appropriate to measure levels of the herbicide along with the lipophilic peptides for the samples in this study.

  1. Authors thank Reviewer 1 for the remarks. Nevertheless, the aim of the study was the evaluation of the chemical component profile of haemolymph from P. Melas exposed to PND. However, the persistence, bioaccumulation, toxicity, and potential for long-range transport of PND are reported in literature (for example JOURNAL OF TOXICOLOGY AND ENVIRONMENTAL HEALTH, PART B 2017, VOL. 20, NO. 1, 1–21, http://dx.doi.org/10.1080/10937404.2016.1222320).

  1. The authors speculate epigenetic effects of exposure of the beetles to PND. Is this the case because of the known downstream action of the identified neuropeptides or is this just a general case of effects not involving genome mutations?

  1. It is clear that PND is toxic and carcinogenic agent, nevertheless other epigenetic phenomena may also play a role in PND-induced toxicity. Moreover, PND-induced genetic mutations are never been described.

Reviewer 2 Report

The work presented for review, entitled:MS-based profiling of haemolymph from Pterostichus melas (P. Melas) exposed to pendimethalin herbicide contains the results of MS analyses of differences in peptide composition in individuals exposed to the herbicide. In my opinion, the work contains minor errors that should be corrected:

1. The title of the work is not very clear. I suggest changing it to: "Mass spectrometry based peptide profiling of haemolymph from Pterostichus melas (P. Melas) exposed to pendimethalin herbicide."

2. I suggest making changes to the abstract to make it more understandable to a potential audience. Abbreviations appear in it that are not explained. It would be appropriate to provide an elaboration of them.

3. Figures 3, 4 and 5 are unreadable. The quality of the figures should be improved.

4.  Were the mass spectra calibrated? There is no information about this in the materials and methods section.

Author Response

Reviewer 2

The work presented for review, entitled: “MS-based profiling of haemolymph from Pterostichus melas (P. Melas) exposed to pendimethalin herbicide” contains the results of MS analyses of differences in peptide composition in individuals exposed to the herbicide. In my opinion, the work contains minor errors that should be corrected:

The title of the work is not very clear. I suggest changing it to: "Mass spectrometry based peptide profiling of haemolymph from Pterostichus melas (P. Melas) exposed to pendimethalin herbicide."

response: The title of the work was modified as follow: "Mass spectrometry-based peptide profiling of haemolymph from Pterostichus melas exposed to pendimethalin herbicide."

I suggest making changes to the abstract to make it more understandable to a potential audience. Abbreviations appear in it that are not explained. It would be appropriate to provide an elaboration of them.

response: The abstract was modified as suggested by the Reviewer.

Figures 3, 4 and 5 are unreadable. response: The quality of the figures should be improved. High quality figures were uploaded in the revision procedure.

Were the mass spectra calibrated? There is no information about this in the materials and methods section.

response: P 13, L 434-436 The sentence “MS analysis ………………….. of low molecular weight species.” was modified as follows “MS analysis was performed in positive reflectron mode, mass accuracy of 10 ppm and a mass range 900-5000 Da were set to record the untargeted fingerprint profile of low molecular weight species. The peptide mass standards kit (Calibration Mixture 1, AB SCIEX) was used to calibrate the MALDI TOF/TOF mass spectrometer. “